# Airbnb's Negative Externalities from the Consumer's Perspective: How the Effects Influence the Booking Intention of Potential Guests

**Chaang-Iuan Ho \*, Tzong-Shyuan Chen and Chin-Pei Li**

Department of Leisure Services Management, Chaoyang University of Technology, 168 Jifeng E. Rd., Wufeng District, Taichung 413310, Taiwan
\* Correspondence: ciho@cyut.edu.tw

**Abstract:** Local governments worldwide have been making efforts to regulate Airbnb and its negative externalities (NEs), as peer-to-peer (P2P) accommodations have grown exponentially. This study seeks to explore the perceptions of potential guests regarding the NEs of Airbnb accommodation by using a contextual approach and multiple methodologies to understand the effects of their choice. Through an experiment involving the collection of data on the responses of 296 participants from Taiwan's post-Millennials and ordered probit model estimations, this study provides a quantitative analysis to distinguish the factors and NEs affecting the likelihood of choosing Airbnb. Under the circumstance where concern for NEs was not included, the results indicated that the accommodation environments and interactive experiences were among the significant Airbnb service features that attracted consumers. However, when NE factors were added a potential effect was identified, with the community environment and security assurances being factors that decreased the likelihood of choosing Airbnb. Previous experiences of staying in hotels were also found to reduce the acceptance of Airbnb as an accommodation mode. These research findings provide insights into Airbnb preferences which could assist in improving the administrative and managerial efforts of P2P accommodation platforms.

**Keywords:** perceived negative externalities; environmental concerns; strategic regulation; ordered probit model; Airbnb

## 1. Introduction

A collaborative economy is one in which individuals in an equal relationship exchange goods and services that can be used by those who need them and when they are not used by owners [1]. The development of the internet and information and communications technology (ICT) allows for this type of transaction so that people share access to resources through community-based online services. The rise of sharing economies has resulted in the development of various resource-sharing platforms, such as Airbnb, which allows property owners to provide accommodation in their primary or secondary residences [2]. By identifying the division between the "ownership" and "use" of resources, such a business supply model provides tourists with different alternatives from which to choose their preferred accommodation space at a price that suits their budget, thereby giving them greater flexibility and choice than is normally available through traditional hotel systems [3]. Homeowners (hosts) can generate supplementary revenue without incurring additional costs, and therefore, Airbnb promotes tourism development and economically benefits local communities [4]. This phenomenon has drawn the attention of researchers, and various studies have examined its success and attractiveness that have led tourists to choose this type of accommodation [3,5–12].

On the other hand, critics of the platforms emphasize that P2P for tourists in residential areas also give rise to negative externalities for residents [5,13–16]. Furthermore, there have

been significant discussions about the gentrification of neighborhoods [17] and negative spillover effects, such as damage and even vandalism within the community [18]. NEs are defined as the secondary low-efficiency effects produced during the resource allocation process [19] and have been regarded as hidden risks for guests [4,13]. Airbnb, as the leading international P2P accommodation platform, has transformed from a niche market for accommodation into a significant option for tourists. Several studies have been conducted with regard to how consumers value the different aspects of Airbnb accommodation and have indicated that guests choose P2P platforms with similar incentives and risk factors [5–7,20,21]. Despite considerable research efforts, very little is known about how consumers' perceptions regarding NEs influence their choice of Airbnb accommodation. Recent research has indicated the vital role played by customers' subjective norms in triggering revisit intention [22]. However, the subjective norm in both studies has been found to be the positive external influence rather than the negative one regarding the choice of accommodation. Furthermore, perceived risk with the purchase channel did not have any impact on the intention to book on Airbnb [23]. It is most likely to be the case that consumers may not receive full information regarding Airbnb's NEs even though there is a website where people share their bad experiences and seek to inform others about the risks and dangers of using Airbnb (AirbnbHell.com). Generating insights into this issue remains a challenging task.

That Airbnb triggers gentrification [17,18,24] and the importance of enforcement on P2P accommodation platforms has been emphasized [25–27]. Most traditional regulatory frameworks for accommodation have proved to be incapable of slowing the growth of P2P platforms [28]. P2P accommodation may need to be regulated to internalize the external effects and reduce the negative external effects. In addition, to regulate tourism housing it is necessary to take into consideration the needs of tourists, industry, the environment, and the host communities. The issue regarding the perceptions and attitudes of neighbors (or residents) has attracted attention in academia [4,17]. We must focus on how the consumers themselves confront the same problems and their responses. Palombo [29] believes that the market and self-regulation are insufficient to deal with Airbnb's impacts on neighborhoods. Although the question as to whether gentrification of Airbnb came first remains open [30], the outlook would not be optimistic if consumers were to continue to patronize and choose Airbnb due to a lack of consideration regarding the NEs. In fact, consumers should better dominate the market and determine the quality of the accommodation products because they deserve legal and safe listings. Oskam and Boswijk [31] suggested that Airbnb's evolution may differ between different cities, primarily as a function of consumer demand and regulatory policies. Based on the "user-pay" principle, this is a promising research direction in that it examines whether the factors regarding NEs influence guest choices of Airbnb accommodation and the kind of impact that these factors have on decision making.

Thus, a contextual approach has been adopted that is grounded in the principle of an ecological perspective on environmental psychology [32]. Environmental contexts involve not only the physical environment but also the economic, cultural, and social environments. The multiple contexts affect individual experience and behavior. Winkel, Saegert, and Evans [33] emphasize the importance of adequate representation of the physical and social contexts instead of focusing on single or isolated variables. In addition, new measurement approaches have been developed that allow for the modeling of multiple covarying components. For example, Mocák et al. [34] implement the 15-minute city concept (FMC) in Slovak cities and attempt to reduce inequalities between different parts of cities, which is one of the consequences of poorly regulated suburbanization processes. We echo the advocacy of Winkel et al. [33] and the contextual analysis specifies a set of situational boundary conditions that qualify the relationship among the target predictor (the NEs of Airbnb accommodation) and the response variable (the booking intention of Airbnb accommodation). It is assumed that potential tourists recognize that NEs in the Airbnb accommodation neighborhood context can be described in terms of physical, economic, and social domains. That is, the Airbnb accommodation environment is no longer a direct, one-to-one cor-

respondence with behavior that is uniform across all potential guests. We use multiple methods that consider the issue of interest (Airbnb booking intention), including measuring variables at different scales (the physical and social contexts), using quantitative techniques to evaluate the relative importance of various complex interdependencies among potential guests and Airbnb accommodation settings. The research question is concerned with the modeling process: Do NE factors have differential impacts on the individual's booking intention? Instead of using structural equation modeling, the ordered probit model was applied to estimate the choice propensity of consumers for Airbnb accommodation under different circumstances. This alternative approach is a regression model for analyzing a latent variable, and in the present study, the variable has been defined as "booking propensity of Airbnb accommodation". Thus, we may predict whether the consumer will make a booking or not given some explanatory variables related to that consumer [35]. In addition, this quantitative model can estimate both changing environmental predictor variables and changes in behaviors occasioned by shifting environmental information.

The primary objective of this research is to clarify the underlying relationships between the choice of Airbnb accommodation and other exogenous factors, such as accommodation services, perceptions of Airbnb's NEs, and the choice likelihood. A few studies employ structural equation models, which often use the values of the ordinal data themselves for analyzing purchase intention [21–23]. The booking intention implies the likelihood of consumers choosing Airbnb accommodation. Since the values of the ordinal data only indicate the order of the degrees of likelihood, the numerical values per se are meaningless. It would, however, be meaningful to examine the magnitude of the relation between the ordinal values. Therefore, an ordered probit model is useful for statistically analyzing the data for such ordinal choices. Furthermore, we may predict whether the consumer will make a booking or not given some explanatory variables related to that consumer. We tackle the research questions surrounding Airbnb in a different way by adopting another modeling approach. Previous studies on Airbnb's NEs have tended to employ qualitative analyses [4,13] or have identified the existence of a spillover effect from the popular Airbnb neighborhoods [18]. To the best of our knowledge, the present study is the first to adopt a quantitative approach to reveal the key components of NEs that influence the booking intention of guests in relation to Airbnb accommodation.

The present study fills a research gap by revealing the effects of NEs on guests' decision making concerning Airbnb accommodation. It contributes to the existing literature by providing theoretical insights into this important research issue. The authors also hope that the findings can promote further discussions on the quantification of Airbnb's NEs. The research results, grounded on guests' contextual and subjective decisions, may assist in policy making with regard to the regulation and enforcement of P2P accommodation platforms as well as consumer protection measures. The remainder of this paper is organized as follows. Section 2 briefly reviews the theoretical background and previous literature; Section 3 describes the data collection, and Section 4 explains our methods of data analysis. Section 5 presents the results of the model estimation, followed by the conclusions, managerial implications, and suggestions for future research in Section 6.

## 2. Literature Review

### 2.1. Theoretical Background

#### 2.1.1. Contextual Analysis

Darley and Gilbert [36] have characterized environmental psychology (EP) as a problem-centered rather than theory-centered set of activities for the solution of community problems, especially for understanding the ecological context of behavior and the transactions involving people and places [37]. EP focuses on the bio-psychosocial view of health and illness, which replaces the single-cause and single-effect models with those that address the complex interactions among the physiological, psychological, and social dimensions of well-being [38]. The emphasis on contextual theories in psychology has been regarded as a conceptual shift within the behavioral sciences away from exclusively

intrapersonal explanations of behavior toward those that encompass both the immediate social environment and the broader cultural, historical, and geographic circumstances of people's daily activities [39–42].

Thus, according to Stokols [32], the proposed dimensions for analyzing community interventions include the spatial, temporal, and sociocultural scope of an analysis, the integration of objective and subjective perspectives on environment and behavior, the use of both individual and aggregate levels of analysis, and the partitive or compositive representation of situations. These dimensions provide a framework for developing contextual theories or those that account for the cross-situational variability of psychological and behavioral events. The application of theoretical strategies for mapping the context of behavior serves as a tool for discovering the situational boundaries of psychological phenomena, specifying the dimensions in which diverse settings could be meaningfully compared and estimating the applied utility of the policy recommendations before they are implemented. For example, Matlovičová et al. [43] propose the CPTED (crime prevention through environmental design) concept, which offers a possible approach by enhancing the sense of security in the urban environment.

To address the concerns regarding the impact of NEs on Airbnb accommodation, we focus on a set of issues that are relevant to the following issues: (1) the contextual nature of tourist experience and action within the physical settings of Airbnb accommodation environments; (2) the adoption of methodologies that consider the complexities of the tourist–environment relations in Airbnb accommodation; and (3) the use of data analytic strategies that reflect the contextual nature of tourist–environment relations.

### 2.1.2. Discrete Choice Models

When individuals must select an option from a set of alternatives, discrete choice modeling methods are used to explain or predict whether an alternative will be chosen [44]. The most common theoretical framework for generating discrete choice models is the random utility theory, which postulates that the probability of individuals choosing a given option is a function of their socioeconomic characteristics and the attractiveness of the option [45]. The important properties of the discrete choice model are summarized as follows:

1.  Individuals always select the option which maximizes their net personal utility, subject to legal, social, physical, and budgetary constraints (time and money).
2.  The observable utility is usually defined as a linear combination of variables, where each variable represents an attribute of the option or the individual. Thus, the relative influence of each attribute, in terms of its contribution to the overall preference regarding the option, is given by its coefficient.
3.  It is assumed that the individual's choice set is predetermined; this implies that the effect of the constraints has already been considered and does not affect the process of selection among the available alternatives.
4.  The dependent variable is an unobserved probability (between zero and one) and the observations are the individual choices (which are either zero or one in the binary logit model or the ordinal values in the ordered probit model).

The quantitative model investigating the psychological factors influencing guests' booking intention in the Airbnb context is theoretically grounded in the discrete choice model. The ordered probit modeling approach involves a regression model for analyzing a latent variable [35], and in the present study, the variable has been defined as the "propensity to book Airbnb accommodation". Apart from a brief presentation of the theoretical paradigm, this study introduces and discusses the concept of NEs as an additional constraint to deepen our comprehension of the attribute factors influencing guests' booking intentions regarding Airbnb accommodation in the following sub-section.

*2.2. Negative Externalities in the Airbnb Context*

2.2.1. The Concept of Negative Externalities

The theory of environmental economics defines externalities as the environmental side effects produced during an economic activity, which are often neglected by the producers [46]. When individuals engage in economic behavior, a portion of the benefits derived do not belong to the individuals but benefit those around them, or a portion of the costs incurred as a result are not borne by the individuals and cause those around them to suffer [47]. Therefore, externalities exist when the behavior of a person or a company affects the utility of another person, company, or community.

Laffont [48] claimed that externalities may be positive or negative and that they occur alongside production or consumption activities. Positive externalities bring benefits to others. In the case of Airbnb services, homeowners sharing space with temporary tenants through the home-sharing platform is a form of non-rivalrous consumption [12]. Undeniably, Airbnb provides positive environmental and social benefits to local tourism industries [49]. Lazăr [19] perceived NEs to be the secondary low-efficiency effects produced during the resource allocation process, which often occur when asset rights are indefinite or non-existent. Accordingly, the producers of NEs should take responsibility for the external costs generated as these types of inefficiencies can give rise to uneasiness in the market [50].

2.2.2. Airbnb's Negative Externalities

Airbnb produces a negative externality in terms of its impacts on the host's neighbors. Guttentag [15] argues that the concern relates to zoning laws, and it is understandable for people to not want to live across the hall from what is essentially a hotel room. The NEs of Airbnb affect the general public, particularly communities, and include both direct and indirect effects [51]. The direct effects include the consumption and occupation of community resources [31], the disruption to neighborhood long-term relationships by transient guests, and the use of community public facilities, such as parking lots, elevators, gyms, and swimming pools, in order to extract the maximum benefits from the paid accommodation without adequate management or appropriate regulatory mechanisms.

Noise complaints and security concerns are major causes of conflict between neighbors and Airbnb hosts [13]. Airbnb guests take advantage of their short time in a foreign place by having parties, getting drunk, singing, or chatting with friends late into the night—all of which can disturb the quietude of the neighborhood and increase the noise annoyance [2,3] and the negative impact on community hygiene and sanitation (e.g., an increasing amount of litter, garbage, and air pollution) [13]. Traffic also becomes an issue of concern [51]. Apart from these considerations, Airbnb guests give rise to security concerns [3,13,52], as the constant presence of strangers could threaten the safety of residents. As guests are not registered, the short-term rentals could harbor criminals or become sites where at-risk populations commit crimes [53], which could adversely impact local communities by causing conflict with community residents [3] or result in an increase in crime [4,13]. Another threat relates to fire safety. Hotels have strict fire regulations; however, the lack of emergency exits in Airbnb accommodation could affect guest and community safety [54]. Obviously, Airbnb's NEs affect not only the community residents, but also the consumers themselves.

The impacts of transforming private housing into tourism accommodation have also generated discussions in relation to anti-social behavior [55]. This behavior, which has further negatively impacted guest–tourist relations [14], along with tourism gentrification, rent increases, loss of neighborhood identity, and displacement of commercial businesses, is among the main reasons for the intensification of anti-tourism protests organized by residents [56]. For example, changes in land use have occurred where residential properties have been converted into buildings for commercial use [57]. Convenience stores may be converted into souvenir shops, which would reduce the level of convenience for local residents [55]. The phenomenon reflects the so-called Airbnbfication which the residents endure. In addition, there have been many studies on P2P management and regulation,

covering subjects such as the decline in government tax revenue [3] and the effects on the traditional hotel industry [2,58]. Tourist destinations have been found to encounter less spending, a shrinking hospitality job market, and a reduction in full-time job vacancies [3].

Although Airbnb claims it is committed to protecting both hosts and guests, it also states that disclosing guest information and ensuring customer protection and safety are important considerations [31]. The Airbnb guests, hosts, and community residents are all stakeholders in the property use, residential maintenance, and leasehold interests. Coupled with the privacy and personal information security considerations, there is a gap between the content promised by the online transactions and the actual services provided [59]. The most striking example is the possible disparity in perception and value between the two P2P parties. Potential hosts (or even community residents) and guests face compatibility issues with the use of resources, which means that paid items, such as cleaning and service fees, should be specified. Second, when hosts provide incomplete information, guests may be subject to high transactional risks. For example, there may be a gap between the promised room quality and the actual condition [59]. In addition, there may not be a specific cancellation policy for hosts, and this would encroach upon the rights and interests of the guests when hosts abruptly and unexpectedly terminate agreements [13]. Indeed, distrust has been found to be a key obstacle in Airbnb consumer attitude formation [21].

The problem of Airbnb gentrification in some cities has been discussed [17,18,56,60]. The progressive gentrification carries negative effects and alienates residents. The major impacts of gentrification include rent increases, the displacement of long-term tenants, a shortage of rental property, and the worsening of the life quality of residents [17]. Evidence has been found of increasing living costs and environmental degradation in neighborhoods with problems of gentrification [18]. Although Airbnb cannot be regarded as the initiator of the gentrification process, it should be seen as an intensifier [60]. The demand for more local and authentic consumption experiences by general tourists (not only the wealthy group) has transformed residential areas.

Adamiak [61] argued that Airbnb supply was not a uniform segment of tourist accommodation and its effects on destinations should be considered in relation to territorial context. Indeed, a spare bedroom that is rented out occasionally and a full property that is rented out year-round are highly distinct. Adamiak et al. [62] indicated that Airbnb was active in holiday destinations in Spain, where it often serves as an intermediary for the rental of second or investment homes and apartments. No serious competition can yet be seen between P2P platforms and the traditional commercial tourist accommodation offer in their study areas. In such places, P2P platforms and the traditional hotel sector play a complementary role, which is consistent with the conclusions of previous studies. However, Adamiak [61] found that professional hosts are growing more quickly than P2P hosts. In addition, Gil and Sequera [63] also indicated that Airbnb in Madrid was dominated by professional actors specializing in the business of renting apartments as STRs within the city's Central District, which has generated negative impacts on the economic sustainability of the city and its inhabitants. They argued that the activities of Airbnb inventory owned by hosts operating full-time rentals or professional enterprises did not comply with the principles of the sharing economy. The social interaction with the host was limited or nonexistent and it did not exhibit the local authentic experiences that Airbnb guests seek or the new accommodation form that it emphasizes.

2.2.3. Unpleasant Factors Regarding the Purchase Intentions of Airbnb Guests

Amaro et al. [23] argue that the role of perceived risk has been overlooked in the context of P2P accommodation and found to be insignificant in regard to the consumers' intention to book on Airbnb or their attitude formation [21]. However, Jun [20] concluded that perceived risk negatively affected repurchase intentions and attitudes. Huang et al. [64] explored the factors that have led guests to abandon Airbnb services. Some factors were found to be related to negative externalities, such as perceived policy/regulation bias that disadvantaged guests, misleading listing information, online data management safety/privacy

and/or usability concerns, sanitation issues and/or amenity malfunctions, and safety and security concerns. Thus, it is worth thoroughly examining the effects of NEs associated with Airbnb accommodation on consumer choices.

### 2.3. Choice Factors for Airbnb Accommodation

Some factors related to the choice of Airbnb have been identified, such as cost, social interaction with the hosts and other local people, the opportunity for cultural exchange, unique local experiences, meaningful social encounters, and the ability to obtain insider tips on local attractions [3–7]. Other important influential factors, such as household amenities, homeliness, and large spaces [5] have also been found. Airbnb guests have claimed that sustainability, such as reducing the environmental, social, and economic impacts of their consumption [8], has been a facilitating factor in their accommodation choices. The location factor has been identified as being particularly critical. Tussyadiah and Zach [9] analyzed P2P reviews and found that many guests emphasized quiet neighborhoods within short walking distances of local restaurants and only minutes by bus to the city center. It was also found that P2P accommodation platforms tended to discuss the neighborhoods and local businesses while hotels tended to stress their proximity to attractions [10]. Wang and Nicolau [11] identified 25 explanatory variables in five categories—host attributes, site and property attributes, amenities and services, rental rules, and online review ratings— that had significant effects on the accommodation price listed on Airbnb. As price plays a primary role in accommodation selection, it is expected that these price-determinant variables are highly related to the decision to choose Airbnb. In addition, feeling a sense of belonging and the uniqueness of Airbnb accommodations have been regarded as the key appeals [65].

Some salient P2P accommodation features have been identified compared to traditional accommodation, with unique local experiences [8] and meaningful social encounters [6] found to be the main characteristics that differentiate Airbnb from traditional accommodation services. Guests enjoy experiencing a community-focused, social atmosphere in their accommodation and the ability to make authentic local connections with the assistance of their hosts [5,6]. Mody et al. [12] identified eight Airbnb dimensions—entertainment, education, escapism, aesthetics, serendipity, localness, communities, and personalization— that were found to be superior to those of hotels. Tussyadiah and Zach [9] found that P2P accommodation services were seen to be better at building relationships between hosts and local areas, whereas hotels were seen to provide better functional services and conveniences, such as airport shuttle services, free parking, breakfast options, and in-room services. Belarmino et al. [10] also found that conversations with hosts was a key attribute in guest reviews on P2P accommodation and that room amenities such as food, beverage, and odor were the predominant theme in hotel reviews. Therefore, as in Mohsin and Lengler [7], it was concluded that guests saw Airbnb as an alternative to traditional accommodation. Except for the host profiles, few differences have been found between the reasons for tourist choices regarding Airbnb and those of traditional accommodation, such as B&Bs [66]. The findings of Blal et al. [67] suggest that improvements in Airbnb quality may impact hotels negatively. However, Guttentag and Smith [68] examined guests' comparative performance expectations from their last Airbnb with hypothetical nearby hotels and found that Airbnb was generally expected to outperform budget hotels/motels, underperform upscale hotels, and have mixed outcomes versus mid-range hotels when considering traditional hotel attributes (e.g., cleanliness and comfort). Thus, tourists who have past experiences of staying in hotels may be concerned about Airbnb accommodation.

Apart from the aforementioned findings, other factors regarding consumers' preferences in relation to Airbnb have been identified. Some scholars have looked into the psychological factors motivating tourists' decision making concerning their accommodation choice and have found that the subjective norm plays a critical role in affecting consumers' Airbnb accommodation bookings [22,23]. Four factors—socio-economic, environmental, technological, and media factors—largely influence Generation Y when choosing Airbnb

as their preferred accommodation [69]. In view of these findings, we argue that consumers have this response due to the lack of consideration involving NEs: the environmental factor may not include the related elements [69] and the subjective norm excludes the negative aspects regarding the external influence [22]. A study by Koh and King [70] has indicated that representatives from Singaporean economy/mid-tier hotels and hostels tended to feel that a stricter regulatory environment was needed, even though they did not perceive an immediate threat from Airbnb. With the increasing gentrification brought about by Airbnb, conveying the information in terms of its regulation to the guests of P2P accommodation platforms has become urgently important.

*2.4. Summary*

Based on the aforementioned discussions, there are clearly both positive and negative aspects of the proliferation of Airbnb as a function of the benefits to be accrued from making a purchase as a guest with Airbnb. The following research questions were developed:

5.    Do NEs affect guest choices of Airbnb accommodation?
6.    If NEs affect guest choices, how do they affect the choices of Airbnb accommodation?

**3. Research Methods**

Based on the principle of contextual research, a particular phenomenon (booking Airbnb accommodation) is thought to be influenced by a surrounding set of events (variables). We recognized that the psychological processes are embedded in physical, economic, and social contexts and should define a set of situational or contextual variables that are thought to exert an important influence on the form and occurrence of the target phenomenon. First, an experiment was conducted to investigate accommodation choice by examining the importance of different listing attributes and NEs. Then, this study presented a framework to test the relationship between these factors and the consumers' decision to choose Airbnb. The following 3 hypotheses are proposed:

**Hypothesis 1 (H1).** *There exists a positive relationship between the service attributes for accommodation choice and the consumers' booking intention regarding Airbnb accommodation.*

**Hypothesis 2 (H2).** *There exists a negative relationship between the NEs and the consumers' booking intention regarding Airbnb accommodation.*

**Hypothesis 3 (H3).** *There exists a relationship between the consumers' past experiences of accommodation and their booking intention regarding Airbnb accommodation.*

*3.1. Research Setting, Questionnaire Design, and Data Collection*

To explore the perceptions of potential guests toward Airbnb's NEs and investigate the influence of these effects on their accommodation choices, an experiment was carried out in which the participants were asked to watch a video with relevant images. We referred to the studies regarding consumers' attitudes toward the advertisement to conduct the experiment [71–73]. The film was produced based on media reports from the TV news which were mainly collected by the researchers and represented a range of 5 themes (the residential security of consumers, consumption of community resources, disruption due to neighborhood impacts on the community environment and safety, an increase in real estate prices and a change in land use, and customer privacy and personal information security). The information related to the video is provided in Appendix A. The responses of the experimental participants may illustrate the effect of the film on Airbnb's NEs and respondents' perceptions of the NEs.

The data for the present study were collected using a traditional paper-based survey. The questions were designed to ask about both renting rooms and entire residences. The survey was conducted in 2 stages. In the first stage, the respondents answered questions about their most recent domestic travel and accommodation arrangements and the factors

that they considered important when making their accommodation choices. In the second stage, the respondents watched the video on Airbnb's NEs and then completed the relevant questionnaire items. The detailed procedure for data collection is illustrated in Figure 1.

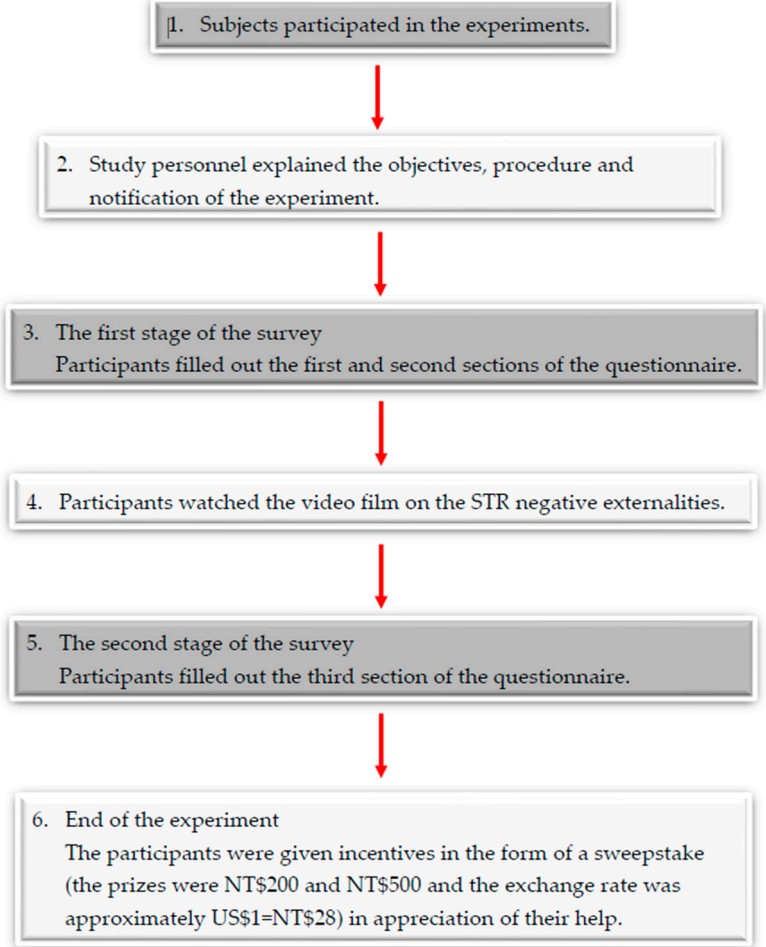

**Figure 1.** Data collection procedure for this study.

Referring to the methods of previous studies [71–73], subjects in this study were undergraduate students. Millennials, often referred to as Generation Y or Digital Natives, are a principal target group for P2P platforms such as Airbnb [22,23]. Indeed, a study by Tussyadiah and Pesonen [8] provided evidence that younger demographics are more prone to the sharing economy. In addition, Rzemieniak et al. [74] indicate that building employer branding in the context of sustainable development is very important for Generation Z (born between 1995 and 2010), also referred to as "post-Millennials" [75]. In other words, these young people may tend to use Airbnb and emphasize the concept of sustainability. Due to the subject matter considered in the study, we have selected the youngest generation, who are currently at the stage of completing their education as students, as a niche group in terms of Airbnb accommodation for our data collection.

The respondents were recruited using a convenience sampling method from the university in which the authors worked in central Taiwan. Permissions and assistance were obtained from faculty colleagues to collect the data in classrooms. Upon agreement to participate, the respondents received information about the research. The subjects were told that the study was designed to obtain college students' reactions to Airbnb. Based on Mitchell [76], they actively processed the information in the video that might have formed/changed their perceptions toward Airbnb. Study personnel facilitated each session, explaining the purpose and format of the session and emphasizing the importance of each participant providing honest responses in relation to the questionnaire.

The questionnaire comprised 3 sections. The first section (7 questions) asked about the most recent domestic travel and accommodation arrangements; the second section (18 questions) asked about the factors that respondents considered when choosing accommodation services, with the degree of importance measured on a 5-point Likert scale from 1 (extremely unimportant) to 5 (extremely important); and the third section had 17 statements about the NEs based on previous research, which were also measured on a 5-point Likert scale from 1 (extremely unimportant) to 5 (extremely important). The respondents watched the video before answering the questions in the third section of the survey. All measurement items or statements were derived from previous studies or online references. To ensure clarity and readability, 22 statements regarding the perceptions of Airbnb's NEs were originally developed and pre-tested by 30 people who had experience of Airbnb accommodation. A criterion was established to eliminate question items that had average scores below 3.0 and standard deviation values greater than 1.5. Finally, 17 statements remained.

Data collection was implemented between 23 February and 15 March 2021, and 296 usable responses were obtained. Thus, the sample was based on students who had Airbnb accommodation experience or were willing to consider Airbnb accommodation in the future but who had not yet used it, or both groups. Given the theoretical rather than applied nature of the present experiment, a student sample was viewed as appropriate [72]. Students are a relevant segment of the population for evaluating the content of the film, and their homogeneity increases the statistical power of the tests of the hypothesized relationships [77].

### 3.2. Dependent Variable: STR Booking Intentions

The last question in the second and third parts of the survey asked respondents about the likelihood of choosing Airbnb accommodation. The responses to these questions were the dependent variables, which were designed in an ordered categorical data form using a scale of 1–5 (highly unlikely, unlikely, neutral, likely, highly likely). The paired-t test result shown in Table 1 indicates that the difference in the likelihood of booking Airbnb accommodation before and after being provided with the related information was statistically significant. Figure 2 illustrates that most respondents changed their minds and were less likely to choose Airbnb accommodation after watching the video on the NEs.

**Table 1.** Respondents' likelihood of choosing Airbnb accommodation W/WO concern for the negative externalities.

| Likelihood of Choosing Airbnb Accommodation | Mean | SD | t-Value |
|---|---|---|---|
| Without concern | 3.584 | 0.7499 | −8.197 |
| With concern | 3.166 | 0.9476 | |

We further analyzed the responses of users and non-users with or without concern for the NEs. The findings indicated that both groups of respondents were less likely to choose Airbnb accommodation after receiving the information related to NEs (see Table 2). In addition, the respondents who had Airbnb accommodation experiences were more likely to repurchase this travel product than those who had never been guests (see Table 3).

**Table 2.** The responses of users and non-users choosing Airbnb accommodation W/WO concern for the negative externalities.

| Airbnb Usage | Likelihood of Choosing Airbnb Accommodation | Mean | SD | t-Value |
|---|---|---|---|---|
| User | Without concern | 4.171 | 0.6177 | −3.100 |
| | With concern | 3.657 | 0.9983 | |
| Non-users | Without concern | 3.506 | 0.7318 | −7.577 |
| | With concern | 3.100 | 0.9228 | |

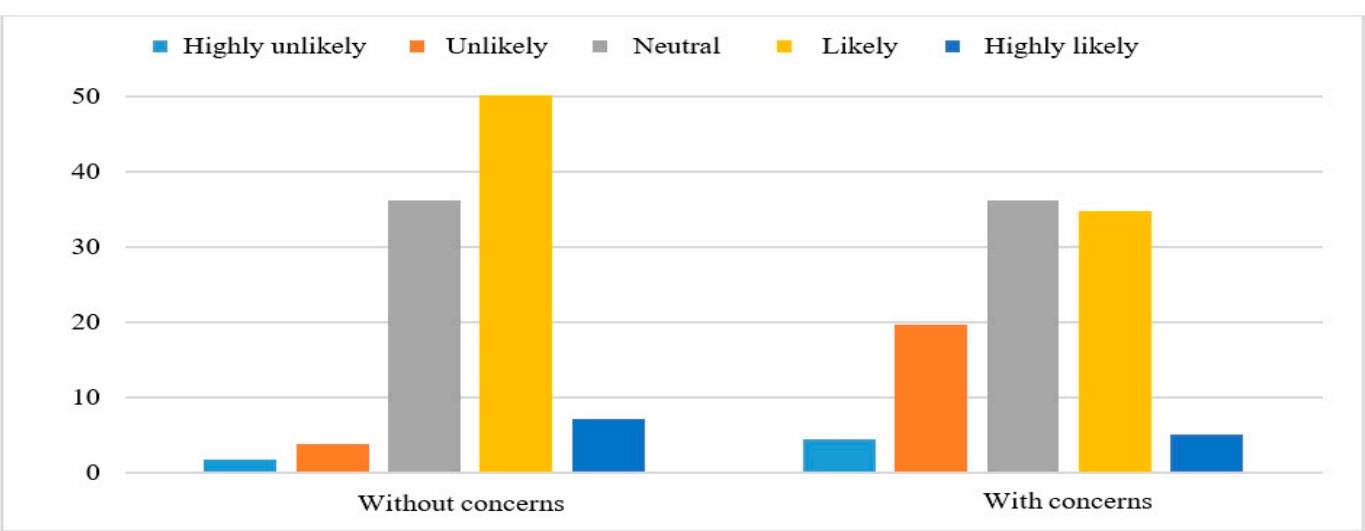

**Figure 2.** Purchase intention of respondents changing W/WO concern for the negative externalities.

**Table 3.** Comparison of the responses of users and non-users choosing Airbnb accommodation W/WO concern for the negative externalities.

| Negative Externalities | Airbnb Usage | N | Mean | SD | t-Value | *p*-Value |
|---|---|---|---|---|---|---|
| Without concern | User | 35 | 4.17 | 0.62 | 5.849 | 0.000 |
| | Non-user | 261 | 3.51 | 0.73 | | |
| With concern | User | 35 | 3.66 | 1.0 | 3.324 | 0.001 |
| | Non-user | 261 | 3.10 | 0.92 | | |

### 3.3. Explanatory Variables

Based on a careful review of previous research on Airbnb accommodation and other accommodation services, the key explanatory variables used in the models were summarized into 3 groups to explain the variations in the likelihood of booking Airbnb accommodation, as follows.

Travel-related characteristics: the first group of variables controlled for the potential impacts of various travel-related characteristics on the likelihood of booking Airbnb accommodation, such as types of domestic travel, travel companions, types of accommodation, in the days before the most recent domestic trip, and the travelers' experiences with various accommodation forms (including Airbnb). From the viewpoint of person–environment congruence, the effects of the physical environment on human behavior vary according to individual characteristics [78]. Thus, in this study, previous accommodation experiences and accommodation preferences were related to booking intention in relation to Airbnb. The information captured by these variables was considered important in explaining the respondents' choices. Table 4 provides the description and the distribution of all relevant variables.

**Table 4.** Respondent profiles for travel and accommodation characteristics (N = 296).

| | |
|---|---|
| **Gender—Respondent gender** | |
| Male | 32.1% |
| Female | 67.9% |
| **Most recent trip [1]** | |
| 0.5 month or less | 40.9% |
| 0.5–1 month | 13.5% |
| 1–3 months | 16.6% |
| 3–6 months | 12.8% |
| 6 months or more | 15.5% |
| **Number of days in the recent trip [2]** | |
| 2-day-1-night | 35.5% |
| 3-day-2-night | 27.4% |
| 4-day-3-night | 5.7% |
| More than 5 days | 3% |
| 1-day | 28% |
| **Accommodation experiences** | |
| Hotel | 90.5% |
| B&B | 89.9% |
| STRs | 11.8% |
| **Accommodation choice for the recent trip** | |
| Hotel | 33.8% |
| B&B | 31.4% |
| STRs | 3% |
| Living in relative/friend's home | 6.1% |
| No accommodation | 25.7% |
| **Traveling companions for the recent trip** | |
| Alone | 1.7% |
| Family members | 34.5% |
| Relatives/Friends | 53.5% |
| Classmates | 6.1% |
| Others | 4.1% |
| **Type of travel [3]** | |
| Self-arranged | 89.8% |
| Package by travel agency | 1.7% |
| Arrangement by travel agent | 2.0% |
| Incentive travel | 6.1% |

Note: [1], [2], and [3] represent missing data and accounted for 0.7%, 0.3%, and 0.3%, respectively.

*Factors for accommodation choice:* Factors for accommodation choice:This group of variables tested the impact of the respondents' preferences for various accommodation-related service attributes and the factors that influenced their choice of Airbnb accommodation. An exploratory factor analysis (EFA) was conducted to delineate the factors underlying the observed variables (the importance of the service attributes included in the survey), and the factor scores were computed from the original service attribute variables. Unlike regression factor scores, Bartlett factor scores are unbiased estimations of the true scores for the corresponding factors and (through orthogonal rotation) are uncorrelated with the true scores for the non-corresponding factors [79]. Table 5 presents the factor analysis results for the attribute factors. A total of 5 factors were retrieved: accommodation environment, interactive experience, hospitality service, geographic location, and cost effectiveness and safety. The results were similar to the findings of Wang and Nicolau [11] for the pricing factors of Airbnb accommodation, which indicated that these factors could reflect the P2P accommodation situation.

**Table 5.** Factor analysis of the service attributes for the respondents' accommodation choice.

| Attributes | Factors and Related Loadings | | | | | Mean | SD |
| | AF1 | AF2 | AF3 | AF4 | AF5 | | |
|---|---|---|---|---|---|---|---|
| A9 | 0.796 | | | | | 3.922 | 0.904 |
| A15 | 0.752 | | | | | 3.956 | 0.833 |
| A12 | 0.738 | | | | | 3.916 | 0.877 |
| A11 | 0.697 | | | | | 4.351 | 0.744 |
| A17 | | 0.810 | | | | 2.642 | 1.022 |
| A8 | | 0.761 | | | | 3.372 | 0.959 |
| A16 | | 0.709 | | | | 3.105 | 0.956 |
| A3 | | 0.518 | | | | 3.696 | 0.929 |
| A1 | | | 0.738 | | | 3.828 | 0.864 |
| A2 | | | 0.662 | | | 4.247 | 0.748 |
| A5 | | | | 0.782 | | 3.784 | 0.902 |
| A4 | | | | 0.768 | | 4.392 | 0.719 |
| A18 | | | | 0.617 | | 4.226 | 0.931 |
| A7 | | | | | 0.711 | 4.236 | 0.806 |
| A13 | | | | | 0.652 | 4.628 | 0.672 |
| A14 | | | | | 0.631 | 4.334 | 0.827 |
| A10 | | | | | 0.620 | 4.797 | 0.527 |
| A6 | | | | | 0.600 | 4.334 | 0.759 |
| Eigen values | 5.169 | 2.043 | 1.172 | 1.123 | 1.025 | | |
| Cronbach's α | 0.754 | 0.720 | 0.514 | 0.613 | 0.697 | | |
| Variance explained (%) | 28.719 | 11.348 | 6.512 | 6.239 | 5.693 | | |
| Accumulated variance explained (%) | 28.719 | 40.067 | 46.579 | 52.818 | 58.511 | | |
| KMO | | | 0.845 | | | | |

Note: Please refer to the Appendix B for the full names of the service attributes represented. AF1 = accommodation environment; AF2 = interactive experience; AF3 = hospitality services; AF4 = geographic location; AF5 = cost effectiveness and safety. Mean = the average score for all indices included in this measure; SD = standard deviation.

*Perception of Airbnb's NEs:* Perception of Airbnb's NEs: Respondents were asked to assess the importance of each statement and the degree to which it affected their choice of Airbnb accommodation. Then, an EFA was conducted on these statements from which 3 factors were retrieved: security assurance, community environment, and information disclosure (see Table 6). In comparison with previous research results [3,4,13], the factors for Airbnb NEs in Taiwan essentially reflected the global concerns.

The subsequent implementation of t-tests, as presented in Table 7, revealed that users exhibited equally favorable (or unfavorable) attitudes towards factors for accommodation choice and perceptions of Airbnb NEs as non-users.

**Table 6.** Factor analysis for the respondents' perceptions of the negative externalities.

| Statements | Factors and Related Loadings | | | Mean | SD |
| | BF1 | BF2 | BF3 | | |
|---|---|---|---|---|---|
| B14 | 0.771 | | | 4.557 | 0.716 |
| B15 | 0.718 | | | 4.554 | 0.753 |
| B7 | 0.682 | | | 4.669 | 0.609 |
| B10 | 0.672 | | | 4.520 | 0.653 |
| B12 | 0.634 | | | 4.118 | 0.930 |
| B8 | 0.631 | | | 4.115 | 0.932 |
| B5 | | 0.761 | | 4.226 | 0.806 |
| B2 | | 0.753 | | 4.253 | 0.737 |
| B6 | | 0.695 | | 3.953 | 0.878 |
| B1 | | 0.657 | | 3.791 | 0.761 |
| B3 | | 0.653 | | 4.500 | 0.638 |
| B4 | | 0.630 | | 3.868 | 0.887 |

**Table 6.** *Cont.*

| Statements | Factors and Related Loadings | | | Mean | SD |
| --- | --- | --- | --- | --- | --- |
| | BF1 | BF2 | BF3 | | |
| B9 | | 0.589 | | 4.220 | 0.841 |
| B16 | | | 0.772 | 4.368 | 0.729 |
| B17 | | | 0.744 | 4.125 | 0.940 |
| B11 | | | 0.671 | 4.361 | 0.816 |
| B13 | | | 0.642 | 4.584 | 0.674 |
| Eigen values | 5.366 | 2.002 | 1.232 | | |
| Cronbach's α | 0.771 | 0.802 | 0.718 | | |
| Variance explained (%) | 31.566 | 11.774 | 7.247 | | |
| Accumulated variance explained (%) | 31.566 | 43.340 | 50.588 | | |
| KMO | | | 0.862 | | |

Note: Please refer to Appendix B for the meanings of these abbreviations represented. BF1 = security assurance; BF2 = community environment; BF3 = information disclosure. Mean = the average score for all indices included in this measure; SD = standard deviation.

**Table 7.** Comparison of the responses of users and non-users of factors for accommodation choice and perceptions of negative externalities.

| Factors | Airbnb Usage | N | Mean | SD | t-Value | *p*-Value |
| --- | --- | --- | --- | --- | --- | --- |
| Accommodation environment | User | 35 | 4.07 | 0.63 | 0.346 | 0.730 |
| | Non-users | 261 | 4.03 | 0.64 | | |
| Interactive experience | User | 35 | 3.11 | 0.79 | −0.788 | 0.431 |
| | Non-users | 261 | 3.22 | 0.70 | | |
| Hospitality service | User | 35 | 4.06 | 0.86 | 0.190 | 0.850 |
| | Non-users | 261 | 4.03 | 0.63 | | |
| Geographic location | User | 35 | 4.18 | 0.72 | 0.459 | 0.646 |
| | Non-users | 261 | 4.13 | 0.63 | | |
| Cost effectiveness and safety | User | 35 | 4.49 | 0.66 | 0.249 | 0.805 |
| | Non-users | 261 | 4.46 | 0.46 | | |
| Security assurance | User | 35 | 4.34 | 0.66 | −0.774 | 0.444 |
| | Non-users | 261 | 4.43 | 0.51 | | |
| Community environment | User | 35 | 4.05 | 0.67 | −0.734 | 0.464 |
| | Non-users | 261 | 4.12 | 0.52 | | |
| Information disclosure | User | 35 | 4.23 | 0.61 | −1.413 | 0.159 |
| | Non-users | 261 | 4.38 | 0.58 | | |

## 4. Specification of the Econometric Model

Ordered probit modeling was used to estimate the respondents' likelihood of choosing Airbnb accommodation. This approach has been used previously in tourism research to analyze tourist satisfaction [80], customer satisfaction, and online hotel reviews [81], as well as in the evaluation of tourism destinations [82]. The common strategy in modeling the ordinal responses is to use either ordered probit or ordered logit models that are estimated using maximum likelihood techniques. This modeling approach was derived and developed based on a straightforward extension of the binary probit model [83]. When the dependent variable is discrete and the multiple and values have a natural order, it is appropriate to use an ordered probit estimation.

In the present study, the ordered probit model regarding purchase intentions as ordinal variables was selected as the analytical method. The items were rated on five-point Likert scales ranging from 'disagree strongly' (one) to 'agree strongly' (five). If the OLS regression method is adopted, the data will be handled as cardinal numbers, assuming that purchase intentions follow a normal distribution. Therefore, using an ordered probit

model is a better strategy for estimating the respondents' likelihood of booking Airbnb given various circumstances.

The dependent variable had five values (one to five). First, one was subtracted from all values so that the values were from zero to four. Similar to the binary probit model, the dependent variable was unobserved and is expressed as follows:

$$Yi^* = \beta'Xi + \varepsilon$$

where Yi* is the dependent variable coded as zero, one, two, three, or four, β is the vector for the coefficients, Xi is the vector for the exploratory variables, and ε is the error term, which follows a standard normal distribution N (0,1).

The dependent variable was observed as the likelihood of booking Airbnb accommodation; therefore, the following were assumed:

$$Y = 0 \text{ if } Y^* < k1,$$

$$Y = 1 \text{ if } k1 \leq Y^* < k2,$$

$$Y = 2 \text{ if } k2 \leq Y^* < k3,$$

$$Y = 3 \text{ if } k3 \leq Y^* < k4, Y = 4 \text{ if } k4 \leq Y^*,$$

where k1, k2, k3, and k4 are 'cut points' and k1 < k2 < k3 < k4. The ordered probit model provided thresholds that indicated the inclination toward choice intention of Airbnb accommodation; however, there were no arbitrary assumptions regarding the magnitude of the differences between the dependent variable categories.

The following conditional probabilities (Pr(Y = 0|X), Pr(Y = 1|X), Pr(Y = 2|X), Pr(Y = 3|X) and Pr(Y = 4|X)) which resulted from the normal distribution can be written as:

$$Pr(Y = 0|X) = Pr(X\beta + \varepsilon < k1) = Pr(\varepsilon < -X\beta + k1) = F(-X\beta + k1),$$

$$Pr(Y = 4|X) = Pr(X\beta + \varepsilon > k4) = Pr(\varepsilon > -X\beta + k4) = 1 - F(-X\beta + k4),$$

$$Pr(Y = 2|X) = Pr(k1 \leq X\beta + \varepsilon < k2) = F(-X\beta + k2) - F(-X\beta + k1),$$

and so on, where F is the cumulative distribution function of residual ε, which is normally distributed N (0,1).

A maximum likelihood procedure was then employed to obtain the results. Ordered probit models have two sets of parameters. The constant and threshold parameters denote the range of the normal distribution associated with specific values of explanatory variables. The remaining parameters represent the effects of the changes in each explanatory variable on the underlying scale and denote the relative importance of each variable in determining the likelihood of booking Airbnb accommodation. Based on Greene [84], one should be cautious when interpreting the signs of coefficients in ordered probit models; they may have different effects on the probabilities of the ordered categories.

## 5. Results and Discussions

The main objective of this modeling effort was to determine whether NEs impacted respondents' choices and to determine the NE factors that were considered most important. The model also investigated the accommodation attributes and the effect of their magnitudes on the propensity to book an Airbnb. Based on the results of the factor analysis, all values significant for every latent variable obtained from the factor analyses were aggregated into variables. After that, models were constructed with the dependent variable being the booking intention in regard to Airbnb accommodation and the independent variables were obtained from the factor analysis plus the previous accommodation experiences of the respondents. To answer the two research questions and test the three hypotheses

regarding the relationship between the booking intention for Airbnb accommodation and the independent variables, three ordered probit models were constructed.

Model 1. Dependent variable: booking intention concerning Airbnb accommodation. Independent variables: based on factors related to accommodation service attributes and previous accommodation experiences (including both hotels and Airbnb).

Model 2. Dependent variable: booking intention concerning Airbnb accommodation. Independent variables: based on factors related to accommodation service attributes, factors regarding NE perceptions, and previous accommodation experiences (including both hotels and Airbnb).

Model 3. Dependent variable: rate of change in booking Airbnb accommodation. Independent variables: based on factors related to accommodation service attributes, factors regarding NE perceptions, and previous accommodation experiences (including both hotels and Airbnb).

All model estimation results are shown in Table 8. Table 9 shows the results of the Wade test, which was used to test the joint significance of several regression coefficients.

**Table 8.** Results of the ordered probit estimation for the likelihood of booking Airbnb accommodation.

| Variables | Model 1 | | | Model 2 | | | Model 3 | | |
|---|---|---|---|---|---|---|---|---|---|
| | Coeff. | Std. Error | Z Value | Coeff. | Std. Error | Z Value | Coeff. | Std. Error | Z Value |
| AF1 | 0.019 | 0.074 | 0.26 | 0.130 | 0.073 | 1.78 * | 0.159 | 0.074 | 2.16 ** |
| AF2 | 0.080 | 0.068 | 1.17 | 0.208 | 0.066 | 3.15 ** | 0.141 | 0.066 | 2.14 ** |
| AF3 | −0.025 | 0.067 | −0.37 | −0.017 | 0.066 | −0.26 | 0.032 | 0.067 | 0.48 |
| AF4 | 0.104 | 0.070 | 1.47 | −0.027 | 0.068 | −0.39 | −0.104 | 0.068 | −1.52 |
| AF5 | 0.068 | 0.070 | 0.97 | 0.012 | 0.073 | 0.17 | 0.015 | 0.073 | 0.2 |
| BF1 | | | | −0.008 | 0.077 | −0.11 | −0.129 | 0.077 | −1.68 * |
| BF2 | | | | −0.205 | 0.069 | −2.97 ** | −0.199 | 0.070 | −2.86 ** |
| BF3 | | | | 0.085 | 0.068 | 1.24 | 0.063 | 0.069 | 0.91 |
| Hotel EXP | 0.099 | 0.224 | 0.44 | −0.268 | 0.216 | −1.24 | −0.403 | 0.218 | −1.85 * |
| STRs EXP | 1.146 | 0.217 | 5.28 ** | 0.716 | 0.199 | 3.59 ** | −0.138 | 0.195 | −0.71 |
| Threshold Constant | | | | | | | | | |
| /cut1 | −2.005 | 0.271 | | −1.995 | 0.239 | | −2.720 | 0.289 | |
| /cut2 | −1.477 | 0.235 | | −0.916 | 0.213 | | −1.632 | 0.223 | |
| /cut3 | −0.023 | 0.215 | | 0.118 | 0.209 | | −0.645 | 0.213 | |
| /cut4 | 1.840 | 0.243 | | 1.611 | 0.232 | | 0.911 | 0.215 | |
| /cut5 | | | | | | | 2.446 | 0.377 | |
| Log likelihood at convergence | | −304.0 | | | −377.8 | | | −359.2 | |
| No. of observations | | 259 | | | 259 | | | 259 | |

Note: ** indicates that the values are significant at the 5% level; * indicates that the values are significant at the 10% level. AF1 = accommodation environment; AF2 = interactive experiences; AF3 = hospitality services; AF4 = geographic location; AF5 = cost effectiveness and safety. BF1 = security assurance; BF2 = community environment; BF3 = information disclosure. Hotel EXP = accommodation experiences (one if respondents had lodged in the hotel and zero elsewhere); STRs EXP = STR experiences (one if respondents had lodged in the STRs and zero elsewhere).

**Table 9.** The results of the Wade test for the joint significance of several regression coefficients.

| Variables | Model 1 | Model 2 | Model 3 |
|---|---|---|---|
| ACCOM attributes | 6.85 | 16.75 ** | 13.34 ** |
| Negative externalities | | 10.09 ** | 14.02 ** |

Note: ** indicates that the values are significant at the 5% level. ACCOM attributes include AF1, AF2, AF3, AF4, and AF5. Negative externalities include BF1, BF2, and BF3.

## 5.1. Model 1 Estimation Results

Before the NE factors were included, the Model 1 estimation results showed that previous Airbnb lodging experience positively affected repurchase intentions. No accom-

modation service attribute factors—accommodation environment, interactive experience, hospitality service, geographic location, and cost effectiveness and safety—had a significant effect on the choice of Airbnb accommodation, and neither the Z-values for these five variables nor the Wade test value (6.85) were significant. These results indicated that the five accommodation service attribute factors commonly considered by young tourists did not play a role in their choice—that is, when only the general accommodation factors were considered, the Airbnb accommodation did not have any particular appeal to the respondents, as previous Airbnb lodging experiences played the key role.

Interestingly, the above-mentioned results may be contrary to prior research that suggested that some accommodation features such as pricing and authentic experiences, were best for a positive experience of Airbnb guests [4]. Admittedly, the respondents in this study had significant leisure, tourism, and hospitality experiences, and so it was not surprising that they were not particularly interested in Airbnb accommodation. They perceived more benefits provided by conventional hotels than guests. Furthermore, because many youth hostels and B&Bs provide some similar services to Airbnb accommodation, Airbnb accommodation was not the only choice. The demand for Airbnb accommodation has been found to be associated with personal accommodation motivations, such as the pursuit of novelty and identification with the Airbnb consumption concepts [8,85]. When these accommodation factors were examined against the push–pull factor framework proposed by Dann [86], the travel accommodation factors were the pulling force and the tourists' personal motivations were the pushing force; therefore, when there are alternatives available in the accommodation market, the consumers' motivations may become the variable dominating their purchase decision-making process.

*5.2. Model 2 Estimation Results*

Model 2 presents the results of when the NE factors were included in the estimation process. The results showed that previous Airbnb accommodation experiences still had a positive influence on the decision to choose Airbnb accommodation, with the two accommodation service variables being positively significant. Respondents who had in the past favored the accommodation environment and interactive experiences were more likely to choose Airbnb accommodation, which was in agreement with the findings of previous studies [3,6,8,20] that found that these accommodation service characteristics of Airbnb were appealing to tourists. Therefore, we could summarize by stating that these two variables plus previous experience of Airbnb accommodation might be perceived as the affective aspect of the respondents' preference for Airbnb accommodation, and the respondents' understanding of NEs could be regarded as the rational aspect. Unlike the finding of Kim and Kim [6], where past experiences played a moderating role in Airbnb's booking intention, this study revealed that this variable had a direct influence on the intention to stay in Airbnb accommodation.

It was found that the effect of the community environment decreased the likelihood of choosing Airbnb accommodation, which indicated that the greater the emphasis on the community environment, the less likely respondents were to choose Airbnb accommodation, which was similar to the findings of So et al. [21] which showed that consumers' perceived insecurity regarding Airbnb accommodation undermined their purchase-related responses. However, the other two variables related to negative externalities had no significant effects on the choice of Airbnb accommodation. While fire safety is unquestionably a critical spillover effect of Airbnb accommodation [54], it may be easily overlooked by guests. A possible interpretation could be that Airbnb properties are often located in residential areas and sometimes guests even share the same residence with the hosts. Residential properties are often not subject to the same rigorous fire prevention and escape regulations that apply to regular hotels, and guests may not perceive the risks associated with P2P accommodation. The two findings of this study were in line with the research results of Amaro et al. [23] with regard to the insignificant impact of perceived risk on the consumers' intention to book on Airbnb. Although these findings may have resulted due to Millennial respondents'

characteristics [23], it is likely that Airbnb guests did not obtain full information regarding the NEs and their regulation, so they did not recognize that such risks were involved. In the context of P2P accommodation, concerns go beyond the issues of safety regarding staying with strangers [3] or privacy and making payment issues online [23].

On the other hand, it was found that the variable for information disclosure did not have an impact on the choice of Airbnb accommodation. Due to the amount of information disclosed by hosts on Airbnb, it is easier for guests to evaluate their intentions [87] and to trust their hosts [88]. Furthermore, the more information provided by hosts, the more likely they are to receive positive reviews, as guests are less likely to be disappointed [89]. Therefore, because of this positive interaction and exchange cycle, this variable may not have a discernable influence on decision making in regard to Airbnb accommodation.

*5.3. Model 3 Estimation Results*

To further understand whether the negative spillover effects had any influence on changing accommodation choice behavior, the changes in the likelihood of the respondents' choice of Airbnb accommodation were analyzed in Model 3. As shown in Table 5, after the factors of negative externalities were considered, the variables for the accommodation environment and interactive experiences had positive coefficients and were significant, which indicated that these two factors were seen as attractive accommodation service attributes of Airbnb. The other three variables—community environment, security assurance, and previous hotel lodging experience—had negative significance, which indicated that these three factors were the rational aspects that that changed the respondents' minds about choosing Airbnb accommodation. This research result was in line with the findings of Agapitou et al. [5], which revealed a trade-off between pricing and security, safety, cleanliness, and additional services for conventional hotels and Airbnb. It indicated that consumers may think more rationally about their choice of Airbnb accommodation when made aware of the NEs. Therefore, by providing information regarding the negative spillover effects of Airbnb accommodation, traditional accommodation operators could shape consumers' perceptions and decrease their likelihood of booking illegal Airbnb listings.

## 6. Conclusions

Airbnb NEs have already become a matter of public concern around the world. Research on regulatory instruments, particularly from the consumer's perspective, remains relatively unexplored despite their importance in managing new rental products. This represents a relevant issue when thinking about its relationship with the regulatory and legal mechanism, especially in terms of anticipating and avoiding the exclusionary effects of NEs already seen. The present study focuses on the consumers' perspective, making it not only different but also easier to link the issues with regard to regulations. It is necessary to reconceive the discussions in moving toward an understanding of how consumers evaluate both positive and negative effects of Airbnb as part of the changes in their decision making. This also allows for contributions from an alternative quantitative approach—ordered probit modeling—that differs from those that predominantly use structural equation modeling (SEM).

The research results confirm the view that Airbnb NEs would decrease consumers' interest in booking Airbnb accommodation and that certain NE factors would have a significant impact on consumer choices. Ordered probit model estimations present the likelihood of booking Airbnb accommodation in certain circumstances. The findings revealed that previous hotel lodging experiences and the NEs in terms of the community environment and security assurance adversely affected the respondents' interest in booking Airbnb accommodation, while the accommodation environment and interactive experience positively affected the choice of Airbnb accommodation. However, not all the factors turned out to be significantly dependent on the consumers' booking intention.

This means that Hypothesis H1 (there exists a positive relationship between service attributes for accommodation choice and consumers' booking intention in regard to Airbnb

accommodation) was confirmed for some aspects. Hypothesis H2 (there exists a negative relationship between the NEs and consumers' booking intention in regard to Airbnb accommodation) was also confirmed, but only in certain respects. Hypothesis H3 (there exists a relationship between consumers' past experiences of accommodation and their booking intention in regard to Airbnb accommodation) was confirmed under the following circumstances: past experiences of Airbnb accommodation had a positive impact on the re-purchase intentions of consumers, but past experiences of hotel stays had a negative impact on consumers' intention to book on Airbnb. Guttentag and Smith [90] found that the recent adopters of Airbnb corresponded with a greater likelihood of using midrange and upscale hotels. Non-adopters exhibited lower novelty-seeking tendencies and innovativeness towards information technology in addition to lower socio-economic status. Compared to these research findings, our research results reveal some similarities and provide some insights into the complexities of the relationships between the accommodation attributes and the NEs. Furthermore, these findings fill a gap in the current literature.

### 6.1. Theoretical Implications

This study is an initial exploration of the effect of NEs on consumers' choice of Airbnb accommodation. It contributes to the related literature by performing a quantitative analysis that provides new insights into the NE factors influencing the booking intention of potential customers in the Airbnb context. By acknowledging the possible risks associated with P2P accommodation, this study illustrates the explanatory power of the model from the consumer perspective. Our research findings add a new layer to understanding the Airbnb accommodation choice and the various factors, including NEs, that influence such a choice. Despite these contributions, however, a few important questions remain unanswered. These include, for example, how Airbnb consumers' attitudes toward NEs are formed and shaped (i.e., shift over time), how residents perceive NEs and the factors that influence their attitudes, how NEs can be transformed in light of the regulations, and how much has been charged by way of permit fees. Pricing is a key issue in a shared accommodation economy [11]. Taking NEs into account and considering them as part of the cost structure for P2P accommodation could be an interesting research issue. As Airbnb and its regulatory environment are evolving rapidly, these questions should be recognized and examined in a timely manner.

The questions raised by Airbnb's business model including housing, taxation, consumer protection, regulation, and liability have been intertwined and integrated. Dann et al. [91] argued that there did not exist a one-size-fits-all approach for handling STRs. Despite individual and novel regulation being better suited to govern Airbnb [31], the business model relies on developments in consumer awareness in terms of more ethical consumption. Existing studies are more focused on discussing the positive or negative impacts of Airbnb rather than on explaining both aspects. In addressing these research gaps, this study proposes preliminary results from the perspective of the Z generation, in view of it being the most critical group of potential Airbnb guests. It may progress into theoretical debates to understand the NEs of Airbnb and the related implications from the consumers' perspective. The main contribution is to enrich the discussions on recent trends in Airbnb regulation, especially considering the exclusionary NE effects already observed in consumer decision making.

### 6.2. Managerial Implications

This study reveals the consumers' requirements for both regular accommodation and Airbnb accommodation. The results offer new considerations for consumers regarding their behavioral obligations between fun and sustainability when enjoying certain resources. As the prices of Airbnb accommodation often exclude external costs, consumers need to understand that Airbnb accommodation does not necessarily consist of low-cost options, which could compel hosts to fulfil their legal obligation and may be more effective than law enforcement. As Avdimiotis and Poulaki [92] stated, Airbnb regulations should have a

structure that is integrated with the aims of tourism regions. The research findings also provide some insights that could be useful to policymakers as they look to manage the Airbnb phenomenon more effectively, as well as to the Airbnb and other P2P accommodation platforms as they look to attract and retain hosts and guests, and other lodging providers as they seek to better cater for tourist preferences. Therefore, in the following sub-section a marketing-driven mechanism is proposed to bridge the various regulatory methods.

### 6.2.1. Recommendations for the Public Sector

Governments worldwide have imposed restrictions to regulate Airbnb NEs [4,29,93,94]. Using regulation to mitigate negative impacts rather than prohibit the service may shape Airbnb more as a maverick accommodation and less of a traditional form [15,16]. Our research results reveal that respondents reacted and cared about the spillover effect when advised regarding the NEs. Policymaking may seek to highlight the benefits of regulations to extend the reinforcement from the consumer perspective. Oskam and Boswijk [31] suggested that Airbnb's evolution differs between different cities, being primarily a function of consumer demand and regulatory policies. Hong and Lee [95] found that South Korea federal government officials were more open than local ones to adapting regulations in a manner favorable to Airbnb, whereas the local officials were more beholden to their local constituents. As Ferreri and Sanyal [25] and Leshinsky and Schatz [26] indicated, the enforcement issue is particularly important because it can be so challenging. Indeed, a regulation should consider dedicated enforcement by being financially supported from the permit fees, and commercial operators should be distinguished from other hosts [27]. Based on Sovani and Jayawardena [96], governments may carefully revise relevant laws and regulations and guide the lodging industry (especially small and medium sized hotels and B&Bs) in order to launch technological innovation and promote healthy competition due to the sharing economy services.

Since the public holds mixed opinions toward Airbnb, a more in-depth understanding of NEs on the part of consumers may lead to a consensus view to craft suitable regulatory controls, accept an ordinance that permits and taxes STRs, and pay the related fees due to the increased rents. Our research results provide a basis to pave the way for their continued acceptance. Regulatory mechanisms and processes for incorporating Airbnb's activities serve as a public issue. Once individuals accept the related concepts and reach a consensus, these concepts may be successfully transferred into a framework of norms, values, and structures for legitimation [97].

Furthermore, governments may adopt marketing strategies to promote regulations and related policies, for example, by making microfilms to educate consumers about the risk they are taking behind the low price or spreading the content of legislations such as the requirement of P2P accommodation that hosts obtain business licenses and have appropriate insurance. Providing detailed information on NEs would be most effective in discouraging consumers from choosing illegal listings. As the costs associated with these taxation and insurance requirements would affect the P2P accommodation prices, this would enhance the safety of guests and reduce the unfair competition being faced by traditional hotel operators.

### 6.2.2. Recommendations for P2P Accommodation Platforms

It was found that Airbnb had positioned itself using alternative notions of community, sustainability, and governance, with a particular focus on individual economic empowerment coming from transformed meanings of "home" while facing the regulatory debate in New York City [98]. Despite Airbnb considering itself to be an intermediary, the issues regarding its self-regulation and enforcement of consumer protection could be further emphasized. Nieuwland and van Melik [4] concluded that the managers of Airbnb and other similar platforms need to take greater responsibility for the impacts of their operations rather than transfer full responsibility to the host for observing local laws applicable to P2P accommodation. As the platform users are charged service fees, platform operators

need to provide better services to their hosts and guests and strengthen the supervision and management of hosts to protect guests [59]. Since these platforms are similar to other accommodation agencies, they must guarantee the quality of their housing products and services. Therefore, platform operators need to specify the host's responsibilities to reduce negative customer experiences [99] and lessen the distrust in the platform [100].

Ferreri and Sanyal [25] called for Airbnb to be involved in the rule enforcement process in view of the difficulties in identifying and collecting evidence on Airbnb violators. Some tasks that should currently be conducted by local governments include the responsibility of platform operators [14], such as filtering and eliminating unqualified hosts, assisting with law enforcement, reconciling disputes, supervising safety checks, and collecting taxes from hosts (and submitting taxes to the government). Leshinsky and Schatz [26] noted that questions may be raised if Airbnb is involved in in the enforcement of public regulations. Nevertheless, Airbnb can play an active role in selecting the listings which meet the criteria and request that the hosts follow the game rules.

The issue regarding full-time vacation rentals with absentee hosts from other STR properties is undoubtedly many regulators' biggest concern, and Airbnb must sort this issue out to some degree. Guttentag [15] argued that it is naive to think that Airbnb will readily remove illegal accommodations because many of the listings removed have quickly been re-listed by their hosts and bulk removals of professional hosts have only occurred in a few places and only following intense pressure and scrutiny. This makes it difficult to regulate the listings as a whole. Restrictions on entire homes/apartments, multiple listings, or renting at least 90 days per year may be considered and executed. Nevertheless, consumer concerns and actions may push Airbnb to better cooperate directly with local governments and become more proactive in its efforts to limit major violations rather than waiting for greater enforcement of regulations against hosts. For example, sharing data can be used to monitor both impacts and regulatory violation, thereby guiding listings with regards to multi-unit operators and appealing for self-regulation to comply with local ordinances.

Midgett et al. [101] argued that Airbnb should typically be more sustainable than hotels, including the aspects of energy use, emissions, water use, waste production, economic well-being of users, and the creation of social ties. Certainly, participation in collaborative consumption is generally expected to be highly ecologically sustainable [1]. For sustainable consumption, consumers and Airbnb hosts cannot ignore the existence of NEs. These may become additional criteria for the listings as, for example, the hosts may be required to meet at least one of the above requirements. Thus, the brand image of the platform may be enhanced.

### 6.2.3. Recommendations for P2P Accommodation Hosts

As Guttentag [94] indicated, managing Airbnb and other P2P accommodation platforms is always a great challenge because the rentals are easy for hosts to establish and difficult to accurately monitor. Negative external effects expose consumers to risks. Some studies have indicated that perceived risk has negative effects on Airbnb users' intention to stay [6,7,20]. The context of NEs is highly related to guest concerns about the situation, such as the accommodation causing them to experience discomfort, anxiety, and tension. Therefore, Airbnb hosts must consider consumer concerns regarding NEs in order to respond to them. It is necessary to enhance accommodation security and safety policies by taking actions such as implementing effective security features and emergency lines and services [7]. RFID key fobs could be a choice for Airbnb hosts looking to maintain the security of their property and provide a convenient experience for their guests [102,103]. In addition, hosts should lessen the physical risk exposure by reducing environmental harm and recycling resources to enhance sustainable value for guests [7].

Furthermore, it is necessary to provide adequate information to facilitate decision making in accommodation choices. As host distrust has been identified as a factor [21], accommodation platforms must provide more meaningful information to guests, such as

meaningful reviews [104]. Guttentag [15] argued that self-regulatory feedback mechanisms are more effective than traditional government regulatory regimes. For example, information accessibility replaces the need for traditional licensing, and companies share data with regulators to help prevent and respond to problems. The real-time data generated by consumers present some advantages over the information that can be gathered by an inspector hired by the government. The number of customer reviews has a strong negative influence on price [105]. The user reviews may serve as a regulatory mechanism which exhibits accommodation weaknesses, such as fire safety or the presence of carbon monoxide detectors, and these issues are considered by government regulations. Thus, review systems are better suited to complementing and bolstering traditional regulatory practices than providing a substitute for traditional regulations. Encouraging Airbnb guests to post information regarding the surrounding area and neighborhood may complement government monitoring and regulatory processes and further support the host and the platform's reputations.

Guttentag [16] pointed out that the combination of an STR monitoring mechanism and more aggressive legal maneuvering may allow destinations to better manage Airbnb properties. Indeed, in doing so, Airbnb properties will inevitably become a more traditional and accepted feature of communities around the world and may eventually be accepted as a more traditional segment of the tourism lodging sector. The local authenticity that Airbnb guests seek is supposed to be delivered by their hosts. It is believed that this is critical for the competition between P2P accommodation platforms and the traditional tourism lodging sector. This may cater to tourist demand for memorable and transformative experiences deep within host communities without involving the social disruption that exerts a negative impact on the quality of life of residents.

### 6.2.4. Recommendations for Other Accommodation Operators

Although the development of Airbnb and other similar platforms has impacted low-cost hotels [13], Forgacs and Dimanche [2] indicated that hotels have ample opportunities to compete with Airbnb on these same grounds—by competing on value rather than rates, enhancing their websites (in a more intuitive and user-friendly way), incorporating local elements into their properties, and embracing customer relationship management practices. A recent study by Richards et al. [17] revealed that the authentic experience promised by P2P accommodation platforms was illusory from the resident's perspective. They argued that gentrification and displacement will impact tourist experiences as they become significantly over-represented, and the opportunities for host interaction become limited. In addition, Sthapit et al. [59] found that the three components of negative memorable Airbnb experiences were dirty and poor room conditions, bad and rude host behavior, and poor customer service, which implied that some Airbnb hosts were unable to provide hotel-grade professional services. It was determined in their study that P2P accommodation shares overlapping customer bases with youth hostels and has similar accommodation characteristics to B&Bs. Indeed, Admiak et al. [62] pointed out that Airbnb has been targeted by different customers, in that, they compete against hotels and other accommodation services of lower categories, but not against high-end hotels. Therefore, traditional accommodation has a competitive advantage because it has fewer negative externalities. When NEs become internalized and P2P accommodation prices increase accordingly, consumers may reconsider the advantages provided by traditional accommodation, such as hospitality services and guaranteed quality assurance. These advantages for traditional accommodation operators could be stressed in marketing.

The study participants (residents) in Richards et al. [17] perceived that Airbnb was not a promoter of cultural exchange but rather a business opportunity for hosts, a low-cost option for guests, and a contributor to the gentrification of the city of Barcelona. Tourists and residents were found to have different schedules that rarely coincided with each other. They classed Airbnb as merely a rental agency for a handful of operators, mainly sublets where the owners were absent and, in most cases, unknown. This leads to speculation

regarding the relatively safe and registered B&B services which are inherently characterized by guest–host interactions [66] and provide higher levels of privacy and security than P2P accommodation. These factors, along with lower prices, are the additional competitive advantages for registered B&B establishments to compete with P2P accommodation. Therefore, in addition to superior accommodation environments, B&Bs could emphasize the host and guest interactions and the B&B architecture, as well as offering unique experiences by leveraging and organizing sightseeing resources in neighboring areas. Through these efforts, B&B operators could maintain their market share.

*6.3. Limitations and Suggestions for Future Research*

This study has several limitations. First, the respondents were local university students and did not include any people from other age groups or different socioeconomic backgrounds. In addition, they were reliant on subject information regarding NEs of Airbnb provided by this study. Therefore, the results cannot be generalized. Future research could survey a wider group of respondents from a range of backgrounds and age groups and use comparative analyses with samples from other regions/countries to extend this study's results.

Future studies could also test the push–pull factor effects of tourist motivations on the choice of Airbnb accommodation considering that only the pull factors (i.e., accommodation attributes) were analyzed in this study. Tourist motivations, which serve as push forces, have previously been found to be significant in the choice of Airbnb accommodation [21,85]. Further investigations on young millennials, who often associate traveling with self-enhancement and self-transcendent experiences [106], are also needed to re-assess whether these variables, as well as accommodation attributes and Nes, can better reveal the various factors affecting guest preferences.

As some of Airbnb's NEs are rarely observed in Taiwan, such as host–guest discrimination [13], poor or no definition of the rights and obligations between hosts and guests, and the impact of Airbnb on community neighborhoods [55], these issues were not included in the analysis. Therefore, these NEs need further exploration. One area for future research is to develop better measures of perceptions of NEs, which may capture different aspects of person–environment transactions. This requires a more thoughtful and systematic approach in identifying and measuring the salient constructs of NEs that have differential impacts on individual behavior. This study has explored the relationship between microeconomic factors and the presence of Airbnb as most of the academic literature has. Therefore, it would be of great interest for future studies to add significant macroeconomic variables and exogenous variables linked with the socioeconomic environment of each district/region/country. This would be important to deepen understanding regarding the macro–micro relationship in the implementation of Airbnb regulation.

Furthermore, future studies may also focus on the complexity of the setting towards which NE policy recommendations are targeted. This would entail considering not only the physical settings but also the economic, cultural, and social contexts that may affect individual experiences and behaviors in Airbnb accommodation environments. As Serrano, Sianes, and Ariza-Montes [107] pointed out, better policies and more adapted strategies may be implemented by local governments and the tourism industry by understanding and identifying the different models implemented in each territory. There is also a need for developing new measurement approaches that allow for the modeling of the multiple covarying components in complex environments. Furukawa and Onuki [108] adopted a quantitative measure referred to as STR friendliness to draw hypotheses on the relationships between STR regulations and relevant socioeconomic indicators. Finally, despite the development of the pricing models for Airbnb listings [11,105], adding the NE factors could be considered.

**Author Contributions:** Conceptualization, methodology, and writing, C.-I.H.; software, validation, and formal analysis, T.-S.C.; investigation and data curation, C.-P.L. All authors have read and agreed to the published version of the manuscript.

**Funding:** This work was funded by the Ministry of Science and Technology, Taiwan, R.O.C. (grant no. MOST 109-2410-H-324-008).

**Institutional Review Board Statement:** Not applicable.

**Informed Consent Statement:** Informed consent was obtained from all subjects involved in the study.

**Data Availability Statement:** The data presented in this study could be obtained on request from the corresponding author.

**Conflicts of Interest:** The authors have no conflicts of interest.

## Appendix A

The video regarding the Airbnb's negative externalities in this study can be accessed at https://drive.google.com/drive/folders/1tKa8K93U_S4uyJbQNrQur5OhDQIwUMgX?usp=sharing (access on 20 February 2021).

## Appendix B

**Table A1.** Questionnaire items in this research.

| | |
|---|---|
| **Accommodation service attributes** | |
| A1 | Close to site or event attractions |
| A2 | Service staff friendly and polite |
| A3 | Unique accommodation experiences |
| A4 | Provision of meals |
| A5 | Close to scenic area to meet the trip requirements |
| A6 | The value derived for money spent |
| A7 | Guest's word-of-mouth and recommendation |
| A8 | Interaction with B&B host or landlord |
| A9 | Convenient transportation |
| A10 | Cleanliness of rooms |
| A11 | Providing self-catering facilities with good levels of comfort and amenity |
| A12 | Provision of a comfortable ambiance |
| A13 | Located in a safe neighborhood/feeling safe in the room |
| A14 | Security of payment |
| A15 | Having specific architecture or appealing decorative design |
| A16 | Provision of a local trip |
| A17 | Interaction with other guests |
| A18 | Having single/double/twin accommodation available |
| **Statements regarding Airbnb's NEs** | |
| B1 | Airbnb guests may engage in community resource consumption (e.g., use of parking lots, elevators, or other public facilities). |
| B2 | The noise caused by Airbnb guests disturbs community residents. |
| B3 | Criminals may choose to stay in Airbnb accommodation (as hiding locations), which in turn become blind spots for social security. |

**Table A1.** *Cont.*

| | |
|---|---|
| B4 | Airbnb causes a shortage of long-term rentals and a surge in prices in the local region. |
| B5 | Airbnb affects the hygiene and sanitation of the community (e.g., an increased amount of garbage and a need to frequently clean water tanks). |
| B6 | Conflicts may arise between Airbnb guests and community residents. |
| B7 | Airbnb listings must have fire safety facilities and fire escape routes designed in compliance with legislation. |
| B8 | Airbnb hosts should pay relevant taxes. |
| B9 | The presence of Airbnb guests may pose a threat to the safety of community residents. |
| B10 | Airbnb guests must be covered by housing-related insurance (e.g., public liability insurance, fire insurance). |
| B11 | Airbnb hosts should specify the paid items (e.g., cleaning fees or service fees). |
| B12 | A certificate of interior renovation is required for Airbnb listings. |
| B13 | Airbnb hosts should lay down specific cancellation policies. |
| B14 | Flame (fire) proofing must be provided for the carpets, curtains, wallpapers, and decoration materials in Airbnb listings; the use of fire-rated doors and paint is required. |
| B15 | Airbnb listings must be legal (with business licenses and registration certificates). |
| B16 | The actual condition of Airbnb accommodation should fulfill guest expectations. |
| B17 | The information and addresses of Airbnb hosts should be available for their guests. |

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
