# Peer review of "Airbnb’s Negative Externalities from the Consumer’s Perspective: How the Effects Influence the Booking Intention of Potential Guests"

_sustainability, doi:10.3390/su15118695_

Round 1
Reviewer 1 Report
PAPER: How do consumers’ environmental concerns affect booking intention of Airbnb accommodation: The effects of negative externalities
It is a complete manuscript and fits the aims and scope of the journal’s topic.
Therefore, at least a ‘’Minor Revision’’ is required to substantially improve this
manuscript. Specifically, the reviewer has the following comments:
POINT 1: Line 17: of instead on
POINT 2: The introduction is well-written and seems very academic
POINT 3: A check to eliminate some typos is needed (for example, lines 146, 149, etc.)
POINT 4: It is recommended to mention in the introduction and/or conclusion the concept of sharing economy, which enriches the reading by better contextualizing Airbnb Platforms.
POINT 5: Line 151: This reference should be removed (Schwart,1982)
POINT 6: Line 171: The use instead of the utilization
POINT 7: Line 215: Please add the date of publication
POINT 8: The authors should add a reference for this strong sentence: ''The NEs of Airbnb affect the general public and particularly the communities and include both direct and indirect effects.'' LINE 231
POINT 9: Line 348: The point is missing
POINT 10: In my opinion, the authors should use another color in figure 1 (without using the red one) Line 400.
POINT 11: Line 440: This reference should be removed (Burton & Lichtenstein, 1988)
POINT 12: Line 456: Figure 2 should be aligned.
POINT 13: The conclusion is very short; it lacks some information.
POINT 14: Video (Appendix A) Line 900?
However, more citations needed.
E.g.
Line: 156-159: What is the connection between territorial development and accommodation (Airbnb)?
Dávid Lóránt, Bujdosó Zoltán, Patkós Csaba. A turizmus hatásai és jelentősége a területfejlesztésben (The impact and importance of tourism in territorial development)
In: Süli-Zakar, István (szerk.) A terület- és településfejlesztés alapjai (Basics of spatial and settlement development), Budapest, Magyarország, Pécs, Magyarország: Dialóg Campus Kiadó (2003) 471 p. pp. 433-453., 21p.
How can we use the RFID technology in the Airbnb?
Adam Novotny, Lorant David, Hajnalka Csafor
APPLYING RFID TECHNOLOGY IN THE RETAIL INDUSTRY – BENEFITS AND CONCERNS FROM THE CONSUMER’S PERSPECTIVE AMFITEATRU ECONOMIC 17 : 39 pp. 615-631. , 17 p. (2015)
Reviewer 2 Report
I have the following minor comments:
*The topic can be improved.
*The scientific contribution of the study should be clearer.
*The literature should be updated.
*The policy recommendations should be results-based.
*I can see some typo errors.
Reviewer 3 Report
The article is highly original and also makes great contributions to the doctrine, but the most applied part of the data still has some inconsistency, although it shows relevant results. In my opinion, the authors should establish a broader connection between the theory or revision of the doctrine and the applied part of the data.Author Response
Please see the attached file.

Reviewer 4 Report
The study deals with a very interesting topic that is currently widely discussed in the academic community. The area of peer-to-peer (P2P) accommodation has become an issue of importance in recent years, particularly in relation to environmental impacts. Then, to be able to meaningfully regulate the supply of P2P accommodation is to regulate its negative externalities it is necessary to understand the motivations in the consumer choice process. Hence, I think the study is of considerable relevance to planning practice but equally enriches and advances the academic discourse on the topic. I agree with the authors' view that: „ ... there have been significant discussions about the gentrification of neighborhoods [14] and negative spillover effects, such as damage and even vandalism within the community [15]“ The above-mentioned issue of negative effects and the possibilities of their reduction through elements of the urban environment were also addressed: Matlovičová, K; et al., 2016. Environment of estates and crime prevention through urban environment formation and modification. Geographica Pannonica 20 (3) , pp.168-180. Similarly in the case of the claim: „Environmental contexts involve not only the physical environment but also the economic, cultural, and social environments as well. The multiple contexts affect individual experience and behavior. Winkel, Saegert and Evans [30]“, , where similar conclusions were reached by the authors in a study: Mocak, P; et.al. 2022. 15-Minute City Concept as a Sustainable Urban Development Alternative: A Brief Outline of Conceptual Frameworks and Slovak Cities as a Case. Folia Geographica, 64 (1) , pp.69-89.) It is a quality study based on a well-developed and original methodology for obtaining and processing relevant data. The paper has a logical structure, relies on relevant sources and provides an interesting perspective on the undoubtedly widely discussed problem the peer-to-peer (P2P) accommodation regulation. The study is well-balanced in content, uses correct methods and I definitely recommend it for publication after minor changes.
Reviewer 5 Report
The topic is relevant and important for the field of study. The author provided a good introduction and literature review was clearly explained. The theories related were also well-provided.
The author should explain in more details regarding how the data collection was implemented and how to screen and select the respondents to the study.
The results and discussions were fine. The conclusions were clear.
The theoretical contribution should be further discussed on how the findings helped extended the theory or solved some of the limitation of the theory.
Future research directions should be addressed more clearly. In addition, the contributions for the stakeholders, such as home owners or others related to the Airbnb should also be explained.
Overall, the paper is a good research study. Job well done.
Reviewer 6 Report
Please show age groups in the table.
Regarding data collection, please indicate that when it started.
